# Deep Learning on Basal Cell Carcinoma In Vivo Reflectance Confocal Microscopy Data

**DOI:** 10.3390/jpm12091471

**Published:** 2022-09-08

**Authors:** Veronika Shavlokhova, Michael Vollmer, Patrick Gholam, Babak Saravi, Andreas Vollmer, Jürgen Hoffmann, Michael Engel, Christian Freudlsperger

**Affiliations:** 1Department of Oral and Maxillofacial Surgery, University Hospital Heidelberg, 69120 Heidelberg, Germany; 2Department of Dermatology, University Hospital Heidelberg, 69120 Heidelberg, Germany; 3Department of Orthopedics and Trauma Surgery, Medical Centre—Albert-Ludwigs-University of Freiburg, Faculty of Medicine, Albert-Ludwigs-University of Freiburg, 79106 Freiburg, Germany

**Keywords:** BCC, reflectance confocal laser microscopy, deep learning, artificial intelligence

## Abstract

Background: Extended skin malignancies of the head and neck region are among the most common cancer types and are associated with numerous diagnostic and therapeutical problems. The radical resection of skin cancer in the facial area often leads to severe functional and aesthetic impairment, and precise margin assessments can avoid the extensive safety margins. On the other hand, the complete removal of the cancer is essential to minimize the risk of recurrence. Reliable intraoperative assessments of the wound margins could overcome this discrepancy between minimal invasiveness and safety distance in the head and neck region. With the help of reflectance confocal laser microscopy (RCM), cells can be visualized in high resolution intraoperatively. The combination with deep learning and automated algorithms allows an investigator independent and objective interpretation of specific confocal imaging data. Therefore, we aimed to apply a deep learning algorithm to detect malignant areas in images obtained via in vivo confocal microscopy. We investigated basal cell carcinoma (BCC), as one of the most common entities with well-described in vivo RCM diagnostic criteria, within a preliminary feasibility study. Patients and Methods: We included 62 patients with histologically confirmed BCC in the head and neck region. All patients underwent in vivo confocal laser microscope scanning. Approximately 382 images with BCC structures could be obtained, annotated, and proceeded for further deep learning model training. Results: A sensitivity of 46% and a specificity of 85% in detecting BCC regions could be achieved using a convolutional neural network model (“MobileNet”). Conclusion: The preliminary results reveal the potential and limitations of the automated detection of BCC with in vivo RCM. Further studies with a larger number of cases are required to obtain better predictability.

## 1. Introduction

Initial promising study results have paved the way to combine the application of an artificial intelligence with dermatological procedures, and show initially promising results in improving diagnostics and progress in computer-aided surgery [1,2]. The integration of artificial intelligence in the clinical routine encourages further work in the field to minimize human-associated risks and reduce treatment costs [3]. Several workgroups have already showed that an accurate deep learning model for in vivo tissue classification could be achieved with available machine learning techniques [4]. The field of application ranges from the diagnosis of skin cancer lesions to tissue grading in dermatological diagnostics [5,6,7]. In particular, the automated evaluation of confocal microscopic images has proven to be successful in determining the grading of tumor stages [5].

The suitable technologies for in vivo real-time cancer assessments in the head and neck region are, among others: polarization dermoscopy, optical coherence tomography (OCT), confocal laser microscopy (RCM), and multiphoton laser microscopy [8,9,10,11]. It is important to note that nodular lesions share a common clinical appearance but have distinct prognoses. Clinical practice presents a challenge when distinguishing between melanoma, dermal naevi, and basal cell carcinoma [12]. The methodologies listed above are promising noninvasive imaging techniques that may reduce the need for unnecessary biopsies.

In vivo reflectance confocal microscopy (RCM) is a novel and noninvasive technique for tissue imaging. The technology allows a real-time and high-resolution optical analysis of skin lesions, with an estimated sensitivity and specificity of up to 100% [4,13,14,15,16]. The main fields of application are screening, tumor detection, and preoperative tumor mapping [17]. Recently, the technology was included in the guideline for screening and follow-up investigation of malignant melanomas and BCCs [13,18,19]. Due to the characteristics of in vivo confocal scanning, the superficially located histological entities are better representable than tumors deep below the superficial dermis [20]. The main benefit of RCM lies in its ability to investigate normal and pathological skin lesions in vivo with a microscopic resolution. The stacked horizontal imaging technique provides a detailed morphological presentation of the cells. This is an alternative approach to traditional microscopy. Additionally, in vivo RCM allows for rapid imaging of large areas of skin as well as the assessment of lesion borders.

Our work aimed to investigate the feasibility of automated image classification applied intraoperatively, for immediate and direct wound margin assessment. This is especially relevant for the head and neck region, where unnecessarily extensive safety resections may lead to extended functional and aesthetic deficits. On the other hand, there is a risk of recurrence, the need for re-resection, and prolonged hospitalization in cases of incomplete resections of the malignant region. Among all known cancers in the head and neck region, we restricted our preliminary feasibility investigation to BCC in the present study. This was due to several reasons. First, the entity is well-described and easily recognizable in confocal scans, based on the following criteria: Among the common in vivo RCM features of BCC, there is the presence of compact clusters of a uniform population of cancer cells [21,22,23]. Their nuclei appear elongated and are aligned along the same axis, a feature known as “nuclear polarization” [24,25]. Nodular BCC has refractory islets with a line of basaloid nuclei, arranged perpendicular to the axis of the tumor aggregate, leading to the formation of a “peripheral nuclear palisade” [24,26]. These aggregates are occasionally surrounded by dark fissures that well demarcate them from the surrounding stromal tissue [27]. The stroma surrounding the tumor displays highly refractive collagen bundles with ragged margins. The vascularization is also enhanced, and the blood vessels are enlarged [23,28].

In our previous work (currently under review), we demonstrated the feasibility of in vivo confocal scanning directly in the wound margins. 

The current work aims to apply a machine learning model to confocal images of BCC wound margins, investigate its accuracy in tumor detection, and define perspectives and restrictions of this promising diagnostic approach. 

## 2. Materials and Methods

### 2.1. Patients

We conducted a study according to the guidelines of the Declaration of Helsinki (as revised in 2013). The study was approved by the Institutional Ethics Committee of Heidelberg University (reference: S-665/2019). A total of 62 patients presenting with a histologically proven BCC in the head and neck area at the Department of Oral and Maxillofacial Surgery, between September 2019 and November 2020, were taken into account. All patients gave their written informed consent to participate in this study. After pretesting and re-evaluation, a total of 37 stacks with 385 slices were generated from these patients, which were included in the further evaluation and training of the AI. The following data were obtained and saved in the pseudonymized form: age, gender, and information on the histopathological report. Table 1 gives an overview of the patients. Examples of different BCC subtypes and their clinical presentation are shown in the image series below (Figure 1A–C). 

### 2.2. Imaging 

BCCs with histological confirmation were imaged using the RCM Vivascope 3000 (Mavig GmbH, Munich, Germany), according to a standardized protocol. The description of the device and its application for in vivo confocal imaging is presented in detail by other workgroups [17,18,19,20,21,22,23]. Thereafter, conventional histopathological examinations were performed on the samples.

### 2.3. Tissue Annotation

We manually annotated the obtained images by identifying the malignant regions, according to the following criteria:(1)An epidermal lesion accompanied by cellular pleomorphism and damage to the epidermis above the lesion.(2)Tumor cells with monomorphic and elongated nuclei.(3)Nuclei are aligned along a single axis (“nuclear polarization”).(4)A high ratio of dilated blood vessels and leukocytes.(5)Inflammation associated with tumor cells.

The described structures were annotated using 3D Slicer (San Francisco, CA, USA) bioimaging analysis software. Differentiation between different BCC confocal criteria was made using different colors. The annotation process was conducted after loading the high-resolution images into the software. Unclear regions were annotated after discussion with an external confocal imaging expert (Prof. Giovanni Pellacani, Chairman of the Dermatology Department, University of Rome, Italy). Each annotated image was reviewed by two experts trained in in vivo confocal imaging. Cases with a low-quality presentation or artifacts were excluded from training (Figure 2A–C).

### 2.4. Image Preprocessing and Convolutional Neural Network Model

This study employed MobileNet as the convolutional neural network model. As a result of its depthwise separated convolutions, this model has the advantage of reducing processing time due to smaller convolutional kernel sizes. During the training, the weights were adjusted using the Adam optimizer (adaptive moment estimation) and the loss function was the cross-entropy loss. In this study, we trained the model for 20 epochs at a learning rate of 1 × 10^−5^. Data augmentation techniques, such as rotation of ±30 degrees, zooming (80–120%), and horizontal flipping were applied to the training dataset as additional preprocessing steps to combat overfitting. The dimensions of the input images were 256 × 256 pixels. MobileNet models were applied to entire in vivo RCM images in a sliding window manner during the evaluation phase, resulting in pixel-level probability maps for both classes.

By means of the current literature and common practice, five criteria were chosen [29]. Using these criteria (damage to the epidermal layer above the lesion, cellular pleomorphism, elongated basal cells with monomorphic nuclei, nucleation along a single axis (“nuclear polarization”), and an increased number of dilated blood vessels with an increased number of leukocytes and tumor-associated inflammation), we chose the BTI core criteria, since pretests indicated that they were the most frequently occurring and most reliable in terms of validation.

Training and testing of MobileNet was performed utilizing the following approach: in cases where a value of more than 50% of the 256 × 256-pixel area was annotated as cancer, according to the best pretest results, then the whole patch is considered as such and classified as malignant (Figure 3). From 24,448 generated patches, 18,599 were considered healthy and 5849 were deemed to be malignant. For training and testing, these patches were split using k-fold cross-validation to build the training and testing dataset.

### 2.5. Expanding MobileNet and Evaluation on the Validation Dataset 

Using an expanded architecture, we were able to predict an entire heatmap, which was then compared with the annotations of the experts (Figure 4). Expansion of the MobileNet model was performed by taking the output after the convolutional layers (32 × 32 × 512) and performing average pooling with 7.7 kernel and 1.1 stride, with the same padding to conserve the shape of the input images. Hereafter, the fully connected layer was transformed to a convolutional layer with softmax activation with two filters of size 512. The weights in these filters represent the weights from the neurons. The 32 × 32 × 2 image resulting from the process was finally upsampled with a factor of 32. Notably, the in vivo RCM images each had a size of 1000 × 1000. In order to be able to fit the images into the expanded model, the image sizes were increased by mirroring, i.e., the 24 pixels along the entire edge were mirrored over. Further, postprocessing was applied to each heatmap to obtain a segmentation mask:a.Improving the visual appearance of the heatmap by using Gaussian filtering.b.Transforming every pixel with a probability of less than 0.5 to 0 and the rest to 1 using a threshold of 0.5.c.Erosion (which is intended to remove isolated pixels).d.The process of dilation (after the erosion operation, the mask is slightly thinner, and the dilation operation restores this property).

In order to evaluate the performance of the model, the pixel-by-pixel segmentation masks were compared with the ground truth labels (Figure 5, Figure 6, Figure 7 and Figure 8).

### 2.6. Statistical Analysis

We performed the analysis using k-fold cross-validation, with cross-validation parameter k equal to 14. Using this approach, the dataset was divided into 14 sets, and in each training session, 13 sets were used for training and one set for validation. Therefore, each training session had a unique dataset. In order to determine the model’s sensitivity and specificity, testing was conducted using the validation dataset after the training was conducted. Tensorflow and Scikit-Learn, two Python packages, were used for training and evaluation.

The following metrics were considered: sensitivity (refers to the proportion of malignant pixels in the validation dataset that predicted the ground truth malignant pixels correctly) and specificity (refers to the proportion of healthy pixels in the validation dataset that correctly predicted the ground truth healthy pixels). The following metrics were computed on a pixel level by utilizing a confusion matrix: 

The true positive (TP) is a condition where both the predicted mask and the truth mask show a malignant pixel. A false positive (FP) occurs when the predicted mask indicates a malignant pixel, but the actual mask shows a healthy pixel. A true negative (TN) indicates that both the predicted mask and the truth mask contain healthy pixels. A false negative (FN) occurs when the pixel in the predicted mask appears to be healthy, but the pixel in the truth mask appears to be malignant. In order to calculate the sensitivity and specificity, the following formulas were used:$$Sensitivity=\frac{TP}{TP+FN}$$
$$Specificity=\frac{TP}{TP+FP}$$

## 3. Results 

The results from the confusion matrix assessing the pixel-wise masks can be found in Table 2. Our model achieved a sensitivity of 0.46 and specificity of 0.85. In addition to the pixel-wise approach, we evaluated the proportion of correctly predicted ground truth labels in the 1024 × 1024 image stacks. In addition, we calculated the percentage of stacks where the model found the lesions correctly (the model found 40% of the lesions correctly in at least three slices of the entire stack), which was equal to 58%. The percentage of cases where at least one stack was correctly recognized as malignant (using the same criteria as written above) was equal to 78%. 

## 4. Discussion

In the current work, we aimed to evaluate the performance of a deep learning model on in vivo confocal images of BCC. This model was trained on BCC tissues derived from a single institutional department and evaluated with an independent validation dataset generated by k-fold cross-validation. The task of the deep learning model was to recognize at the pixel level whether structures of BCC lesions that were manually annotated by experts in the field of confocal laser microscopy were correctly predicted by the model. The output of the model was a set of heatmaps representing the locations of cancerous and noncancerous regions for tissue classification. Our study significantly contributes to the developments in the field of artificial intelligence-driven diagnostics; to the best of our knowledge, this is the first study to validate the use of a deep learning model on images of basal cell carcinoma, obtained with an in vivo RCM device intraoperatively.

The specificity in detecting and classifying healthy noncancerous regions in our model was as high as 0.85. In contrast, the true positive rate, the sensitivity of the described method, reached only 0.46. This was probably due to the architectural heterogeneity of BCC lesions (different confocal microscopic correlates, the quality of images (grey scales, reflection), localization, and subtyping). In addition, the limited number of cases is another significant limitation, as it leads to overfitting of the model on the training data, affecting reliable accuracy measures on the independent validation dataset. The healthy tissue regions graphically represent clear and similar patterns at the cellular level. Therefore, a higher number of BCC RCM images will be required in order to assess the sensitivity in automatic detection of cancer areas in future studies. This is a well-known problem, also mentioned by other researchers [4]. Considering that imaging databases containing RCM cancer images for deep learning purposes are not readily available yet, multicenter studies or the development of online research databases are of high relevance for the progress in automatized cancer diagnostics.

In this case, as this is a single institutional cohort, the data collected do not represent the true diversity of BCC histological entities. In addition, we only applied binary prediction modelling in our study, which is a simplification of the actual situation and the diversity of the histology. For example, it does not consider gradings or other types of characterization to assess the progress of the malignancy. To verify our results, other independent datasets, which include more feature variables, would be beneficial to expand the presented baseline model and increase its accuracy.

In addition, this technology might be error-prone because it depends on evaluating the images generated by confocal microscopy. Currently, there is no satisfactory solution in the literature to overcome this limitation. In order to be able to work efficiently with confocal laser microscopy, the user needs detailed knowledge and experience in handling and interpreting the image structures [30]. The associated learning curves while establishing such novel techniques will probably be among the most interesting information for the performing surgeons. Structured learning programs by experienced histopathologists and imaging experts could help to overcome this limitation.

This work provides new critical solutions in the field of automatized intraoperative cancer diagnostics by combining a well-known deep learning model, which has been applied now for several years, with confocal laser microscopy. In a study by Aubreville et al. [4], confocal laser endomicroscopy combined with a convolutional neuronal network showed one of the first promising results in detecting oral squamous cell carcinomas, resulting in an area under the curve (AUC) of 0.96. The number of images fed into the convolutional neural network was 11000, translating to a well-trained model.

An essential characteristic of confocal laser microscopy is a low penetration depth into the papillary layer of the dermis, which makes it impossible to adequately measure the depth and thickness of BCC lesions that are particularly extensive [31]. More profound dermal changes such as nodular melanomas, nodular BCC, or panniculitis cannot be fully and therefore adequately imaged by confocal diagnostics. This has been partly solved in current work through application directly in the wound margins and should therefore be considered a great gain of this technique. 

There are various ways to apply deep learning models to images [32]. The main distinction is being made between supervised and unsupervised learning. In order to train a deep learning model through ‘unsupervised learning’, extraordinarily large amounts of data are required, within which the AI learns to recognize structures and regularities on its own [33,34,35]. Since ‘unsupervised learning’ was not appropriate for the number of cases in the present study (14 cases with a total of 37 stacks and 385 slices), the alternative ‘supervised learning’ has been applied in our study. In supervised learning, it is necessary to mark the ground truth on the images before training. In our case, these were the structures defined, which the model would later use to distinguish between the binary class BCC area versus no BCC area. The ‘ground truth’ was manually annotated by the two experts in the field of confocal laser microscopy using the 3D Slicer program. In order to select the structures to be annotated in the best possible way, it was important to find a balance between the confocal microscopic criteria of BCCs, which are frequently and reliably described in the literature, and the process of annotating them (sharpness of the boundary, differentiation between the grey levels at the pixel level for the human eye, time required, reproducibility, etc.) as structures that are essential for deep learning (logical sequence, redundancy, clear boundaries, distinctive features, etc.). This problem has to be mentioned as another limitation of supervised learning [36]. In addition, in contrast to unsupervised learning, supervised learning requires datasets that fulfil a minimum image quality requirement. The large dataset used in the unsupervised learning method forgives quality deficiencies in the confocal images [37]. 

The study’s greatest strength can be seen in the combination of the well-established advantages of in vivo confocal laser microscopy with deep learning. This opens up the possibility for clinicians to apply confocal laser microscopy in the diagnosis and treatment of BCC lesions in the head and neck region and serves as a basis for further investigations.

## 5. Conclusions

In the present study, we were able to demonstrate a specificity of 0.85, even with a small number of patients. We postulate that the automated application of confocal laser microscopy in the head and neck region could be a successful application in future clinical practice. However, we encourage other workgroups to test the provided model with a larger number of cases in order to assess the true positive rate more precisely.

## Figures and Tables

**Figure 1 jpm-12-01471-f001:**
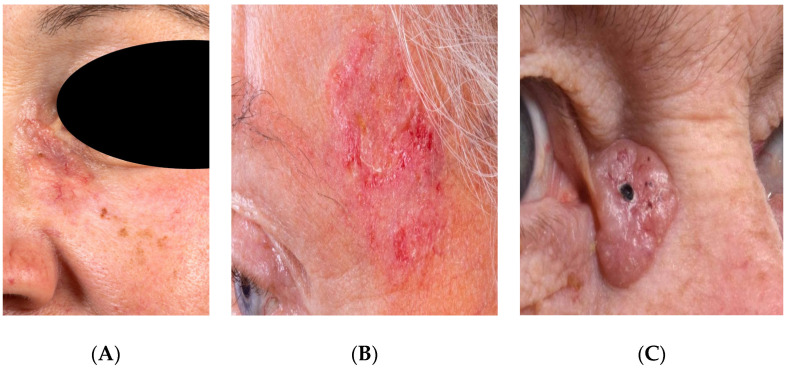
Clinical manifestation of basal cell carcinoma (BCC). (**A**): micronodular, (**B**): infiltrative, and (**C**): nodular histological types.

**Figure 2 jpm-12-01471-f002:**
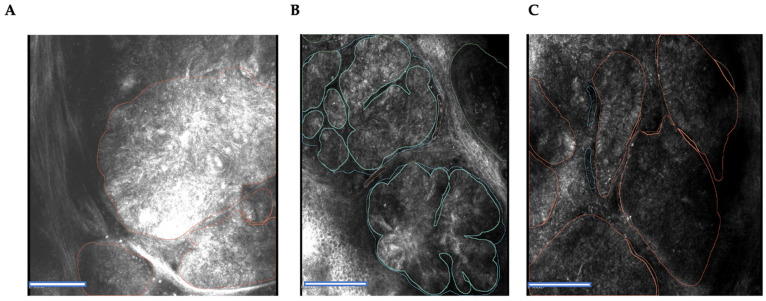
In vivo RCM images of BCC using the example of the criteria of bright tumor islands’ cores (the BTI core in green (**B**) is an example of the criterion, BTI core amorph. The BTI core in red (**A**,**C**) is an example of the criterion, BTI core round). Scale bar, 100 μm.

**Figure 3 jpm-12-01471-f003:**
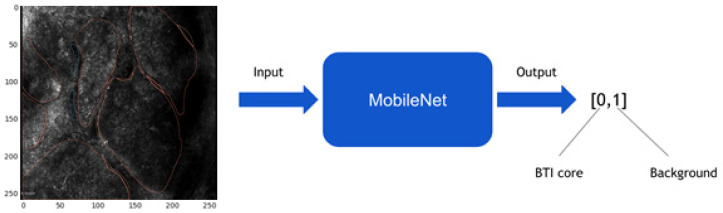
Training MobileNet: less than 50% of the patch area was annotated as malignant.

**Figure 4 jpm-12-01471-f004:**
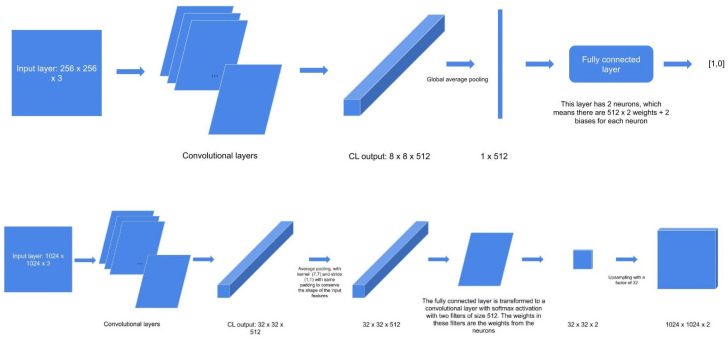
Expansion of the MobileNet model.

**Figure 5 jpm-12-01471-f005:**
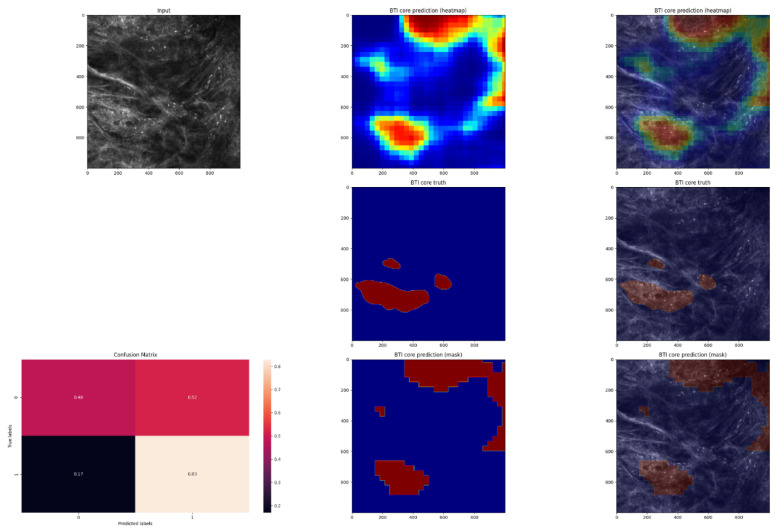
An example of an in vivo RCM image of a BCC that has been manually annotated for cancerous regions and has been predicted by an algorithm. An image of the predicted heatmap can also be seen in the image. Jet is a colormap that ranges from blue to red, where the former represents SCC and the latter, non-SCC. Using the methodology described in the Materials and Methods section, image shows the prediction mask generated from the heatmap. Additionally, a normalized confusion matrix is shown for each row. The class that is not SCC is represented by 0, and 1 represents the class that is SCC. As we can see in this example, this is an average-performing example. The lesions were predicted correctly, although not in their entirety.

**Figure 6 jpm-12-01471-f006:**
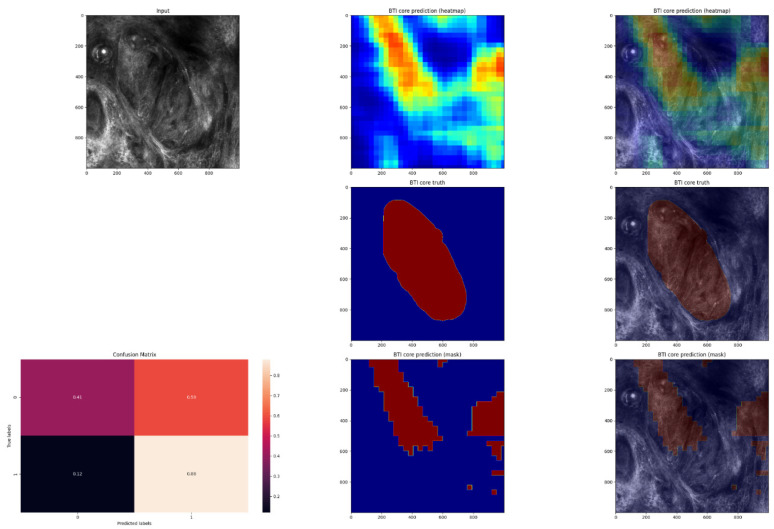
This is the second example of an in vivo RCM image of a BCC, with cancerous regions manually annotated and automated predictions. An image of the predicted heatmap can also be seen in the image. Jet is a colormap that ranges from blue to red, where the former represents SCC and the latter, non-SCC. Using the methodology described in the Materials and Methods section, image shows the prediction mask generated from the heatmap. Additionally, a normalized confusion matrix is shown for each row. The class that is not SCC is represented by 0, and 1 represents the class that is SCC. As we can see, this is an average-performing example. This is one of the better-performing examples.

**Figure 7 jpm-12-01471-f007:**
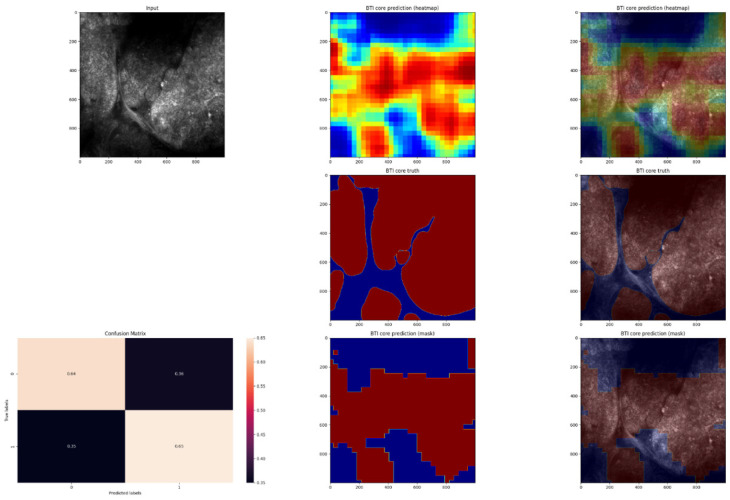
Third example of an in vivo RCM image of a BCC, with cancerous regions manually annotated and automated predictions. An image of the predicted heatmap can also be seen in the image. Jet is a colormap that ranges from blue to red, where the former represents SCC and the latter, non-SCC. Using the methodology described in the Materials and Methods section, image shows the prediction mask generated from the heatmap. Additionally, a normalized confusion matrix is shown for each row. The class that is not SCC is represented by 0 represents, and 1 represents the class that is SCC. As we can see, this is an average-performing example. This is another better-performing example; the majority of the cancerous regions were predicted correctly.

**Figure 8 jpm-12-01471-f008:**
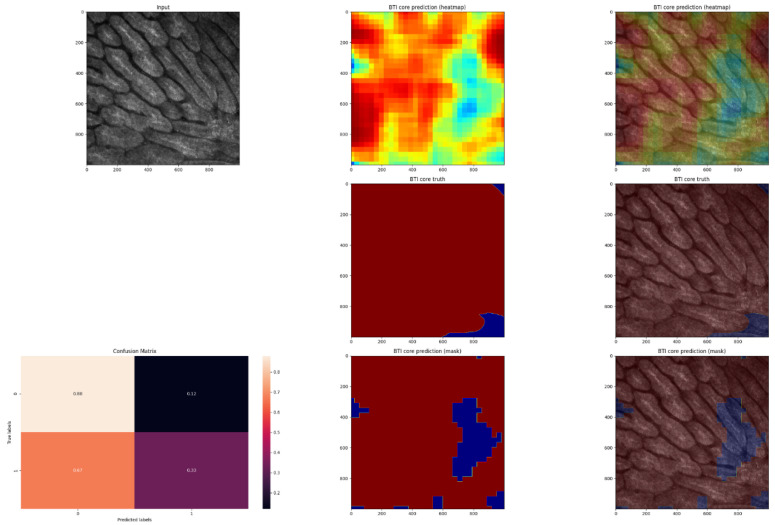
Fourth example of an in vivo RCM image of a BCC, with cancerous regions manually annotated and automated predictions. An image of the predicted heatmap can also be seen in the image. Jet is a colormap that ranges from blue to red, where the former represents SCC and the latter, non-SCC. Using the methodology described in the Materials and Methods section, image shows the prediction mask generated from the heatmap. Additionally, a normalized confusion matrix is shown for each row. The class that is not SCC is represented by, and 1 represents the class that is SCC. As we can see, this is an average-performing example. This is another better-performing example; the majority of the cancerous regions were predicted correctly.

**Table 1 jpm-12-01471-t001:** Baseline characteristics of patients and tumor subtypes.

Sex	Age (Mean)	Subtype	Percentage
male (56%)	71.87	nodular BCC	69%
female (44%)		superficial/plaque/trunk skin BCC	22%
		sclerosing/morpheaform/infiltrative BCC	4%
		infundibulocystic BCC	3%
		pigmented BCC	1%

**Table 2 jpm-12-01471-t002:** Confusion matrix of the testing phase. TN: true negative; FN: false negative; TP: true positive; FP: false positive.

Truth/Prediction	P = Noncancerous	P = Cancerous
T = noncancerous	233,794,335 (TN) 0.85	40,618,083 (FP) 0.15
T = cancerous	58,199,803 (FN) 0.54	49,387,779 (TP) 0.46

## Data Availability

Data supporting the findings of the study are available from the corresponding author upon reasonable request.

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
