# Peer review of "Deep Learning on Basal Cell Carcinoma In Vivo Reflectance Confocal Microscopy Data"

_jpm, 2022, doi:10.3390/jpm12091471_

Round 1
Reviewer 1 Report
This is a very promising study, advancing our knowledge on no-invasive diagnostics of BCC. The Authors very modestly stated that their method (RCM and AI combination) for the time being, has only 46% of sensitivity but clearly stated study limitations and possible ways for improvement. Inevitably the study is a great step forward in non-invasive diagnostic process.
Author Response
Dear Reviewer, thank you very much for your kind judgement on our manuscript and the time you invested.

Reviewer 2 Report
The authors applied deep learning model to confocal laser microscopy images to detect basal cell carcinoma. My main concerns are:
- The structure of the manuscript needs to be improved.
- The sample size is very small to make reliable conclusions.
- Advantages of RCM should be explained in more detail.
Other concerns:
- I haven't seen any studies that used very large and specific k-value in cross validation (k=14). Do the performance metrics change dramatically for different k values? Maybe another experiment could be conducted to analyze effect of k value. Otherwise, I would suggest to keep it small (~3-5).
- In table1, estimated prices for the rows are missing.
- Figure 2, it would be nice to put scale bar in figures.
- Please put parameter values used in data augmentation (zooming, rotation)
- The authors mentioned that "we focused on the criteria BTI-core because it is best for validation". Please give references or justify why it is best for validation.
-"Training a MobileNet model" needs to be rewritten. Difficult to follow.
-"Expanding the MobileNet" section does not explain how the model is expanded. Figure 4 alone is not sufficient.
-Some of the sentences in Results do belong to methodology actually.
-Authors mentioned that the model found the 40% of the lesions correctly in at least 3 slices. Why is it important? Why authors mentioned it specifically?
Author Response
Thank you for your constructive comments. We highly appreciate the time and effort you have invested in our manuscript. We have tried to address each of your recommendations as precisely as possible.

Reviewer 3 Report
The work is well done. But I think that the application of this method could be relegated only in very few specialized centers. But, the bcc is a very common neoplastic lesion with a very large diffusion
Author Response

(The authors gave the same response as above.)

Reviewer 4 Report
The authors performed an interesting study on AI and RCM imaging.
I have some comments:
1) table 1 is too general, it incudes some but not all the commercial producer of the devices, and is out of scope for the title and study aim. I suggest to remove it.
2) Figure 2 should be explained better. Labeling with different colors but these are not explained
3) In the introduction, multiphoton laser microscopy is not mentioned (see: High-resolution imaging of basal cell carcinoma: a comparison between multiphoton microscopy with fluorescence lifetime imaging andreflectance confocal microscopy) and differential diagnosis should be better explained (eg: Nodular skin lesions: correlation of reflectance confocal microscopy and optical coherence tomography features)
4) training dataset and test dataset should be better explained
5) Prof. Pellacani is mentioned in the method section but not in acknowledgments.
Author Response

(The authors gave the same response as above.)

Round 2
Reviewer 2 Report
The authors addressed my concerns. Thank you for the detailed responses and presenting the valuable work.